# Epilepsy-Induced High Affinity Blockade of the Cardiac Sodium Current I_Na_ by Lamotrigine; A Potential for Acquired Arrythmias

**DOI:** 10.3390/ph15101208

**Published:** 2022-09-29

**Authors:** Juan Antonio Contreras Vite, Carlos Vega Valle, Happi Biekeu Mbem, Sarah-Maude Boivin, Robert Dumaine

**Affiliations:** 1Department of Pharmacology and Physiology, Faculty of Medicine and Health Sciences, University of Sherbrooke, Sherbrooke, QC J1H 5N4, Canada; 2Department of Biology, Faculty of Sciences, University of Sherbrooke, Sherbrooke, QC J1K 2R1, Canada

**Keywords:** ion channels, electrophysiology, patch clamp

## Abstract

Lamotrigine is widely prescribed to treat bipolar neurological disorder and epilepsy. It exerts its antiepileptic action by blocking voltage-gated sodium channels in neurons. Recently, the US Food and Drug Administration issued a warning on the use of Lamotrigine after observations of conduction anomalies and Brugada syndrome patterns on the electrocardiograms of epileptic patients treated with the drug. Brugada syndrome and conduction disturbance are both associated with alterations of the cardiac sodium current (I_Na_) kinetics and amplitude. In this study, we used the patch clamp technique on cardiomyocytes from epileptic rats to test the hypothesis that Lamotrigine also blocks I_Na_ in the heart. We found that Lamotrigine inhibited 60% of I_Na_ peak amplitude and reduced cardiac excitability in epileptic rats but had little effect in sham animals. Moreover, Lamotrigine inhibited 67% of I_NaL_ and, more importantly, prolonged the action potential refractory period in epileptic animals. Our results suggest that enhanced affinity of Lamotrigine for I_Na_ may in part explain the clinical phenotypes observed in epileptic patients.

## 1. Introduction

Lamotrigine (Lamictal™, LTG) is an antiepileptic drug approved by the US Food and Drug Administration (FDA) in 1994 to treat bipolar disorders and epilepsy. Part of the antiepileptic action of LTG stems from its ability to block the sodium current [1,2] and possibly the neuronal transient outward potassium current IA [3] expressed in the cortex and hippocampus of the brain. The overall effect is to stabilize the presynaptic membrane of neuronal cells and to attenuate the effect of the excitatory neurotransmitter glutamate during repetitive neuronal firing [4]. Slow binding of LTG during activation or fast inactivation of neuronal sodium channels (nNa_V_s) was proposed to explain the block of sodium current in neurons [2,5,6,7]. Whether LTG also blocks the cardiac sodium current (I_Na_) at therapeutic doses (100–500 mg/day) yielding an effective plasma concentration between 2–16 µg/mL (7.8–62.5 µmol/L) [8,9,10], is unknown. However, clinical observations of Brugada syndrome (BrS) phenotype [11,12,13,14] and QRS prolongation [15] on the electrocardiograms (ECG) of epileptic patients treated with LTG are consistent with conduction disturbances and I_Na_ block. Those observations, coupled with a series of case reports on LTG cardiotoxicity [16,17,18], prompted the FDA to restrict the use of LTG in 2021 and to add a warning label stating that “*Lamictal exhibits class 1B antiarrhythmic activity at therapeutically relevant concentrations.*” [19]. The mechanism by which LTG may exert its arrhythmogenic effect remains unknown. Identifying such potential mechanisms linked to its effect on I_Na_ is the goal of this paper.

In sharp contrast to epileptic patients, studies from toxicology centers indicate that non-epileptic patients on LTG overdose, with serum levels between 17–90 µg/mL, exhibit only minor to moderate neurologic or electrocardiographic effects [18]. This raises the possibility that epilepsy enhances the apparent affinity of LTG for cardiac I_Na_, possibly by enhancing expression of nNavs in the heart. In support, we previously reported that epilepsy increases the contribution of nNa_V_s to I_Na_ [20,21]. Because of the apparent higher affinity of LTG for neuronal cells, we wanted to test if enhanced expression of nNa_V_s in the heart ventricle may contribute to the arrhythmogenic event clinically reported in epileptic patients.

In this study, we present a series of experiments showing that indeed, epilepsy increased the affinity of LTG for I_Na_. The enhanced blockade of I_Na_ also prolonged the action potential refractory period. The results may provide a basis to explain the conduction disturbances and the BrS phenotype associated with LTG toxicity in epileptic patients.

## 2. Results

We first compared the effects of LTG on I_Na_ in ventricular cardiomyocytes from epileptic and sham animals. Figure 1A shows that 10 µmol/L of LTG decreased I_Na_ amplitude more for epileptic animals. Blockade of peak I_Na_ was 16 ± 6% and 61 ± 6% for sham and epileptic animals, respectively. Current voltage relationships (Figure 1B) showed that LTG blockade of I_Na_ by 10 µmol/L and 100 µmol/L was similar in epileptic animals. LTG (10 µmol/L) also shifted voltage dependence of I_Na_ activation in epileptic animals but not in sham. Mid-activation voltage (V_0.5_) was depolarized by 5 mV (Table 1, Figure 1C) in cardiomyocytes from epileptic animals. I_Na_ voltage threshold (I_Na,Th_) was depolarized by 5 ± 1 mV and 10.7 ± 1.4 mV with 10 and 100 µmol/L of LTG, respectively, in epileptic animals (Figure 1D), while I_Na,Th_ was depolarized 8.5 ± 1.5 mV by 100 µmol/L of LTG in sham animals. The maximum Na^+^ conductance (G_Na,max_) decreased by 49 ± 5% in the presence of LTG (10 µmol/L) in epileptic animals but had no effect in the sham group (Figure 1E).

### 2.1. Lamotrigine Blockade of I_Na_

Our results from Figure 1, showing that blockade of I_Na_ by LTG was larger in epileptic animals, combined with our previous data indicating an increased contribution of TTX-sensitive Na^+^ channels [20] during epilepsy, suggest that LTG may target at least two populations of Na^+^ channels in ventricular cardiomyocytes. Accordingly, this should translate into a biphasic dose–response curve for I_Na_. Figure 2A shows that blockade of I_Na_ by LTG in sham animals could be reasonably fit to a single monophasic Hill equation, thus, suggesting that blockade of I_Na_ follows mostly single receptor binding kinetics. In sharp contrast, epileptic rats exhibited a biphasic response to LTG with a high affinity blockade generating a plateau at 47% of the maximum amplitude for concentrations between 10–100 µmol/L and a sensitivity like that of sham animals for larger concentrations. These results indicate at least two LTG binding sites contributed to block I_Na_ during epilepsy. LTG half-maximal blocking concentration (IC_50_) obtained from the fit to data was 155 ± 22 µmol/L in sham animals. In epileptic conditions, IC_50_ values were 211 ± 48 µmol/L and 1.5 ± 0.3 µmol/L (Figure 2B). This result, therefore, indicates an increased contribution of LTG-sensitive channels to I_Na_.

### 2.2. Lamotrigine Prolonged Refractory Period

We next tested if the reduction in I_Na_ caused by LTG could be due to a decrease in Na^+^ channel availability (steady-state inactivation). At therapeutic concentration (10 µmol/L), LTG hyperpolarized mid-inactivation potential (V_h_) of I_Na_ by 2.7 mV and 6.7 mV in sham and epileptic animals, respectively (Figure 3, Table 2). Those shifts in steady-state inactivation would theoretically reduce I_Na_ amplitude by 12% and 18% in sham and epileptic animals when cardiomyocytes are sitting around their natural resting membrane potential of −80 mV and may contribute to a reduction in cell excitability.

I_Na_ recovery from inactivation plays an important role in regulating the minimal time needed between successive action potentials (refractory period). Using standard recovery protocols (Figure 4A), we found that I_Na_ recovery was best described by a sum of two exponentials. Fitting data to a two-exponential function showed that epilepsy alone delayed recovery by increasing the slow recovery time constant by 20%. LTG only slightly slowed recovery of I_Na_ in sham and epilepsy cardiomyocytes (Figure 4B, Table 3).

The refractory period is caused by the time needed for the channels to transit back from the inactivated to the closed state, from which they can re-open. We next tested whether those effects of LTG on I_Na_ recovery translated into a longer refractory period of cardiomyocyte action potentials (Figure 5A,C). In control conditions, the time needed to recover a full amplitude action potential at resting membrane voltages of −80 mV or −100 mV was similar between sham and epileptic animals (Figure 5B,D). LTG (1 µmol/L) increased the total refractory period by 65 ± 11% and 185 ± 13% at −80 mV, and by 106 ± 17% and 173 ± 26% at −100 mV in sham and epileptic animals, respectively. These results indicate that LTG increased the refractory period by a factor of 1.9 for sham and 2.8 for epileptic animals (*p* < 0.001, Tukey test with one-way ANOVA). As previously reported, epileptic rats displayed longer action potentials [20] (Appendix A) at the plateau level in part due to a larger window current (Appendix A). LTG (1 µmol/L) reduced early action potential duration (30% repolarization) by 30.4 ± 4.2% and 33.5 ± 4.2% in sham and epileptic animals, respectively. We next evaluated if the effects of LTG on I_Na_ late current could account for the changes in APD. Our measurements using a voltage ramp protocol showed that epilepsy increased I_NaL_ by 26 ± 9.2%, from −1.5 ± 0.2 to −1.9 ± 0.1 pA/pF (Figure 6A), as we previously reported [20,21]. Following application of 10µmol/L of LTG, I_NaL_ amplitude was reduced from −1.5 ± 0.2 to −1.4 ± 0.2 pA/pF and from −1.9 ± 0.1 to −0.6 ± 0.1 pA/pF in sham and epileptic animals, respectively (Figure 6B). This reduction in I_NaL_ represents a block of 6.7 ± 4.3% in sham and 66.7 ± 5.8% during epilepsy. LTG at 10 µmol/L decreased the window current calculated from the overlap of the activation and steady-state inactivation of I_Na_ by 9.5% and 91.3% in sham and epileptic animals, respectively (Appendix A). These values are close to those found experimentally with our I_NaL_ measurements. Overall, our data show a strong contribution of I_NaL_ blockade to modulation of APD by LTG.

## 3. Discussion

The Na^+^ channel family consists of 10 pore-forming α-subunits associated with auxiliary β-subunits identified as β_1_ through β_4_ [22]. Among them, tetrodotoxin (TTX)-sensitive voltage-gated channels Na_V_1.1, Na_V_1.2, Na_V_1.3 and Na_V_1.6 are the most abundantly expressed in the brain, while the TTX-resistant Na_V_1.5 is considered the cardiac isoform [23]. Overexpression of Na_V_1.1 and Na_V_1.3 within the brain has been reported in non-hereditary forms of epilepsy [24,25]. In a previous study, we demonstrated that epilepsy also enhances expression of TTX-sensitive channels (among them, Na_V_1.1) within the rat cardiac ventricle.

In the heart, sodium channels contribute to conduction and cellular excitability by modulating the voltage threshold, the rising phase, and the duration of action potentials. Although the bulk of sodium current is carried by the cardiac isoform Na_V_1.5, we previously showed that tetrodotoxin TTX-sensitive channels contribute to I_Na_ and I_NaL_ [21,26,27] This contribution increased during epilepsy. We previously reported contributions of ≈19% and ≈35% to I_NaL_ for epileptic rats. Our data (Figure 1) show that blockade of I_Na_ is more important in cardiomyocytes from epileptic animals. Therefore, our observation of a saturating block by LTG is most likely linked to blockade of neuronal channel isoforms overexpressed during epilepsy and contributing to I_Na_. Those data are consistent with our previous RT-PCR and Western blot analyses showing that cDNA and protein expression of neuronal sodium channels, especially Na_V_1.1, is enhanced in cardiac ventricles of epileptic animals [20].

However, we were surprised by the amplitude of the block by LTG during epilepsy. Based on our previous results showing that 35% of peak I_Na_ is generated by TTX-sensitive channels in epileptic animals, we were expecting a proportional level of I_Na_ block by LTG. The 61% block shown on the peak I_Na_-I/V curve and the plateau at 53% block on the dose–response curve for epileptic animals (Figure 2) indicates that some Na_V_1.5 channels might also have been blocked but with a lower affinity, or part of the blockade is due to the inactivation kinetics of I_Na_.

The modulated receptor hypothesis of Hille [28] may explain blockade of neuronal sodium channels by LTG. Under this model, LTG binds to open channels and stabilizes inactivation in a higher affinity block [6,29,30,31,32]. This mechanism is generally thought to be responsible for the use-dependent block of I_Na_ by local anesthetics such as lidocaine [33]. It was proposed to explain the antiepileptic effect of LTG by accumulation of I_Na_ block in the inactivated states during rapid firing in hippocampus [1]. Such a mechanism is consistent with our data showing minimal alterations of steady-state activation in both sham and epileptic animals but a significant hyperpolarization of I_Na_ mid-inactivation potential during epilepsy (Figure 3), as previously reported [2]. The higher affinity block in the inactivated state may also explain the slower recovery of I_Na_ in both sham and epileptic animals. Given that our pulse protocol to measure I_Na_ was 25 ms, nNavs had more time to inactivate, and this may have increased blockade by LTG at lower concentrations. Therefore, the saturating effect of LTG block can be explained by a low-affinity blockade of cardiac Na_V_1.5 channels and a high-affinity blockade of nNavs overexpressed during epilepsy.

A key difference between neuronal and cardiac action potentials is their duration (APD). In neurons from the central nervous system, APD is between 1 and 10 ms [34] as opposed to cardiac ventricular APD, which will vary from 20–80 ms in rodents and up to 450 ms in humans [35]. Therefore, any neuronal sodium channel isoform sensitive to LTG will have a much longer inactivation time in the heart, and this will promote a more important block of I_Na_. Potentiation of I_Na_ blockade may significantly reduce conduction and potentiate ventricular bradycardia by prolonging the refractory period and reducing APD, as observed during our action potential measurements in cardiomyocytes from epileptic animals.

Interestingly, the reduction in APD occurs at 30% and 50% repolarization (Appendix A), which is the region of major I_NaL_ involvement and is more important in epileptic animal vs. sham. In agreement, we found a larger reduction in I_NaL_ during epilepsy. Prolongation of the refractory period is anti-arrhythmic in cases where heart rate is rapid, such as in tachycardia and polymorphic VT. In those type of arrythmias, the beneficial effect comes from slowing the accelerated heart rate or reducing dispersion of electrical repolarization within the ventricle. However, in normal settings, an increase in I_Na_ refractory period could prevent subsequent beats from occurring normally and slow cardiac rhythm to rates that cause bradycardia. In those conditions, slowing recovery of I_Na_ may, therefore, potentiate cardiac arrhythmias. This represents a potential mechanism that may explain the clinical observations of bradycardia and conduction problems in patients treated with LTG.

Late sodium current contributes to AP repolarization plateau, and its enhancement causes a longer QT interval, which triggers arrhythmias. Our results show that epilepsy increased I_NaL_ amplitude by 27%, a value close to what we already reported [20]. Interestingly, LTG at therapeutic concentration (10 µmol/L) reduced I_NaL_ by 66.7 ± 5.8% only in epileptic animals (Figure 6), further suggesting a significant contribution of LTG-sensitive nNavs during epilepsy. Our results show that epilepsy increased blockade of I_NaL_ by LTG by a factor of 10. I_NaL_ inhibition could have benefits against pathologies such as drug-induced or hereditary long QT syndrome [35] by reducing the duration of the ventricular AP duration. We previously reported that epilepsy prolonged ventricular APD by increasing I_NaL_. Our results suggest that LTG may prevent or reduce QT prolongation in epileptic individuals. However, this protective effect may be lost when combined with blockade of the I_Na_ peak (61%). Indeed, a reduction in peak I_Na_ will impact the early phase 1 of the AP by disturbing the balance between I_Na_ and the transient outward potassium current (I_to_), especially in the epicardial layer of the ventricle. The reduction in I_Na_ amplitude will potentiate early repolarization by I_to_. Adding the APD shortening effect of weaker I_NaL_ is likely to further promote early repolarization, a hallmark of BrS. In regions, such as the epicardium of the right ventricle, where I_to_ is most prominent, repolarization will occur rapidly, while the inner part of the ventricle will remain fully depolarized [36]. This dispersion of repolarization within the cardiac ventricle can trigger shunt currents from the still depolarized areas to the fully repolarized regions. Consequently, the fully repolarized regions of the epicardium may re-excite prematurely, thus creating a substrate for re-entrant arrhythmias. This may in part explain the observations of BrS arrhythmias in epileptic patients treated with LTG.

Overall, our data demonstrate that LTG, by virtue of its effect on I_Na_, may cause a reduction in cardiac conduction speed, excitability and APD. Those effects are consistent with clinical observations of QRS widening [18,37,38,39] during LTG overdose and BrS phenotypes [11,12,36,40,41] in epileptic patients. Moreover, because of the enhanced expression of nNavs sensitive to LTG during epilepsy, our data may in part explain why cardiac disturbances seem relatively benign during overdose in patients treated for depression-related illness but may become serious and potentially deadly for epileptic patients. Our study thus provides a basic framework to explain the increased risk of cardiac arrhythmias seen in epileptic patients under treatment with LTG.

## 4. Methods

### 4.1. Animal Model and Cell Dissociation

Epilepsy was induced in rats by subcutaneous injection of Kainic acid (KA, Hello Bio Inc., Princeton, NJ, USA) as previously described [20]. Briefly, adult Sprague Dawley rats (~250–275 g) were injected twice with 8 mg/kg of KA, with a 1.5 to 2 h interval between injections, to induce seizures. The status epilepticus was stopped by intraperitoneal (IP) injection of 25 mg/kg of diazepam (Valium, Rexall Pharmacy Group Ltd., Toronto, ON, Canada) 2 h after KA. Animals were constantly monitored for 36–40 days after induction of epilepsy. Only animals showing chronic seizure behavior consisting of rearing and falling corresponding to stages 4 and 5 on the Racine scale over the course of 36 day were used in this study [42]. Sham animals received diazepam and saline in lieu of KA. Animals were housed 1 per cage on a 14 h/10 h light/dark cycle with free access to tap water and food. All animals were used between 36 and 40 days after treatment. Left ventricular cardiomyocytes from adult rats were isolated by enzymatic dissociation as we previously described [26,43]. Animals were euthanized by exsanguination. Briefly, rats were injected intraperitoneally with 500 U/kg of Sodium Heparin (Sandoz Canada Inc., Boucherville, QC, Canada) 20 min prior to sedation with 1-chloro-2,2,2-trifluoroethyl difluoro methyl ether (Isoflurane, USP, Baxter, Simcoe County, ON, Canada). When the animal was fully anesthetized, the chest was opened, and the heart rapidly excised.

### 4.2. Electrophysiology

I_Na_ was recorded from ventricular cardiomyocytes at room temperature with the patch clamp technique in whole cell configuration using an Axopatch 200B amplifier (Molecular Devices, Sunnyvale, CA, USA). Analysis was performed using the Clampex 10.7 analysis software. Recordings were acquired at 10 kHz and filtered to 5 kHz (lowpass Bessel filter). Whole cell capacitance and series resistance compensation (85%) were optimized to reduce the capacitive artifact and minimize voltage clamp error.

Cells from both sham and epileptic animals were incubated for 10 min with various concentrations of LTG and perfused constantly during patch clamp measurements. I_Na_ recordings took around 20 min. The quality of the giga-seal after the 20 min was, in most cases, not good enough to perform LTG washout and attempt to record the current again.

For action potential (AP) measurements, membrane potential was kept at either −100 mV or −80 mV in current clamp mode (I = 0). Action potentials were triggered by 1 ms pulses of threshold current applied at a frequency of 20 Hz where mentioned. It is important to note that in the LTG experiments, it was necessary to perform the AP measurements with 1 µmol/L instead of 10 µmol/L, because the AP was not triggered in the presence of 10 µmol/L, which introduced substantial variability in the results.

The extracellular recording solution for I_Na_ measurements contained (in mmol/L): 117.5 Choline-Cl, 10 NaOH, 2.8 Na-Acetate, 4 KOH, 1 CaCl_2_, 1.5 MgCl_2_, 20 HEPES, 1 CoCl_2_, 5 TEA, 2 4-AP, 5 BaCl_2_, 10 Glucose. The pH and osmolarity were adjusted to 7.4 (with NaOH) and 295–300 mOsm (sucrose). Pipette solution contained (in mmol/L): 10 NaOH, 5 NaCl, 5 CsF, 2 MgCl_2_, 10 EGTA, 20 HEPES, 120 Cesium-Aspartate, 0.5 GTP, 3 Creatine Phosphate, 2 ATP-Mg. The pH was adjusted with CsOH to 7.3 and osmolarity to 295–300 mOsm.

The solutions for I_NaL_ recordings (in mmol/L) were: 10 NaOH, 5 NaCl, 5 KOH, 4 MgCl_2_, 10 EGTA, 20 HEPES, 120 CsOH, and 4 Na_2_-ATP for pipette solution. The pH was adjusted to 7.3 with CsOH and osmolarity to 290–300 mOsmol/L. Extracellular solution contained: 125 NaCl, 10 NaOH, 2.8 Na-Acetate, 4 KOH, 1 CaCl_2_, 1.5 MgCl_2_, 20 HEPES, 1 CoCl_2_, 5 TEA, 2 4-AP, 5 BaCl_2_, 10 Glucose. The pH and osmolarity were adjusted to 7.4 (with NaOH) and 295–300 mOsmol/L (sucrose).

For AP measurements, the cardiomyocyte bath solution contained (in mmol/L): 126 NaCl, 5.4 KCl, 2 CaCl_2_, 1 MgCl_2_, 20 HEPES, 11 Glucose; and for pipette solution: 90 K-aspartate, 30 KCl, 10 NaCl, 5.5 Glucose, 1 MgCl_2_, 10 EGTA, 4 Na-ATP, 20 HEPES. Extracellular and intracellular solution pH was adjusted to 7.4 (NaOH) and 7.2 (KOH), respectively.

Recording pipettes were pulled from 1.5 mm O.D., 1.16 mm I.D. capillary glass (PGT150T-7.5 Harvard Apparatus) and had resistance between 1 and 3 MΩ.

### 4.3. Materials

Lamotrigine (LTG, ApexBio LLC Technology, Houston, TX, USA) was dissolved in DMSO to make a stock solution of 0.3 mol/L, from which we diluted in extracellular solution to reach the final concentrations. DMSO in LTG extracellular solutions never exceeded 0.2%. No DMSO effects on I_Na_ were observed at this maximum concentration.

### 4.4. Data Analysis

Figures were produced and data analysis performed with the pCLAMP program suite software (Molecular Devices, Sunnyvale, CA, USA) and Origin 8.5 (OriginLab, Nothampton, MA, USA).

I_Na_ current voltage relationships were obtained from recordings during 25 ms voltage steps between −70 and 10 mV, in 5 mV increments, from a −100 mV holding potential. Average peak current values were normalized to membrane capacitance (Cm) and plotted against the test voltage to construct the I_Na_ current-voltage curves (pA/pF). Conductance (G_Na_) was calculated as G_Na_ = I_Na_/(Vm − E_Na_), where Vm is the membrane potential (mV) and E_Na_ is the sodium reversal potential (mV). Maximum conductance (G_Na,Max_) was obtained from the slope of a linear regression fit to the linear portion of the I/V at voltage more positive than −25 mV.

Steady-state inactivation was measured using a 15 ms test pulse to 30mV following a series of 500 ms conditioning potentials from −120 to −40 mV in 5 mV increments. Inactivation curves were obtained by plotting the ratio of I_Na_ to its maximum value (I_Na,Max_) as a function of the conditioning voltage.

Activation and inactivation data were fitted to a standard Boltzmann distribution function: $Y=\frac{\left(A_{1}-A_{2}\right)}{1+\left[\left(Vm-E_{Na}\right)/V_{0.5}\right]}+A_{2}$, where *Y* represents the fraction of activated (m) or available (h) channels obtained, respectively, from the ratio of the macroscopic conductance (G_Na_/G_Na,Max_) or the sodium current I/I_Max_. Vm and E_Na_ were defined as before, and V_0.5_ was the mid-potential for activation or inactivation. G_Na_ was obtained from the current–voltage relationship as G_Na_ = I_Na_/(Vm − E_Na_), and G_Na,Max_ represented the maximal Na^+^ conductance. V_0.5_ was the membrane voltage, where the distribution was half-maximal, corresponding to mid-activation or mid-inactivation (V_h_) voltages.

Recovery from inactivation was measured using a standard double-pulse protocol while holding cells to membrane potentials of −100 mV. I_Na_ was elicited by two-step pulses S1 (100 ms) and S2 (20 ms) to −35 mV, separated by increasing intervals at the holding membrane potential, in increments of 5 ms. Recovery from inactivation was obtained by plotting the ratio of I_Na_ amplitudes (S2/S1) as a function of the time interval ∆t. Time constants (τ) were calculated from a sum of two exponential functions fit to the ratio of I_Na,S2_/I_Na,S1_: $\frac{I_{Na,S2}}{I_{Na,S1}}=I_{0}+I_{s}*exp\left(\frac{-t}{\tau s}\right)+I_{f}*exp\left(\frac{-t}{\tau f}\right)$, where I_s_ and I_f_ and *τ*_s_ and *τ*_f_ represent the fraction of current and time constants for the slow and fast components of I_Na_ recovery, respectively.

I_NaL_ current was induced by applying a depolarizing voltage ramp protocol, from −120 mV to +60 mV, at a 0.04 V/s rate. This protocol inactivated the peak sodium current. Digital subtraction of traces in the absence and presence of 10 µmol/L/L of tetrodotoxin (TTX) yielded the I_NaL_.

*Dose–response curves*. The values of the maximum current density amplitude obtained for each drug concentration were divided by the average of the maximum current density amplitude obtained under control conditions, then the data were plotted as a function of *LTG* concentration tested and fitted to monophasic (sham) or biphasic response (epileptic) Hill absorption isotherm:
$$\begin{array}{c} \frac{I_{Na}}{I_{Na,Max}}=\frac{1}{1+\left(\frac{\left[LTG\right]}{IC_{50,LTGr}}\right)}\left(\mathrm{sham}\right)\\ \frac{I_{Na}}{I_{Na,Max}}=\left\{\frac{f_{LTGrs}}{1+\left(\frac{\left[LTG\right]}{IC_{50,LTGs}}\right)}+\frac{\left(1-f_{LTGs}\right)}{1+\left(\frac{\left[LTG\right]}{IC_{50,LTGr}}\right)}\right\}\left(\mathrm{epileptic}\right) \end{array}$$

Parameters *f_LTG_* and *IC*_50,*LTG*_ represent the fraction blocked by *LTG* and half maximal blocking concentration, respectively. The half blocking concentration of *LTG* was initially determined by fitting the data from sham animals (*IC*_50*LTGr*_) and used as seed value to fit the epileptic animal data to the sum of two Langmuir isotherms, keeping the assumption of a 1:1 binding to all receptors.

### 4.5. Statistics

Experimental data points are presented as data ± SEM, and number of cells is indicated in each Figure. Statistics were performed using the Tukey test comparing groups with one-way ANOVA.

## Figures and Tables

**Figure 1 pharmaceuticals-15-01208-f001:**
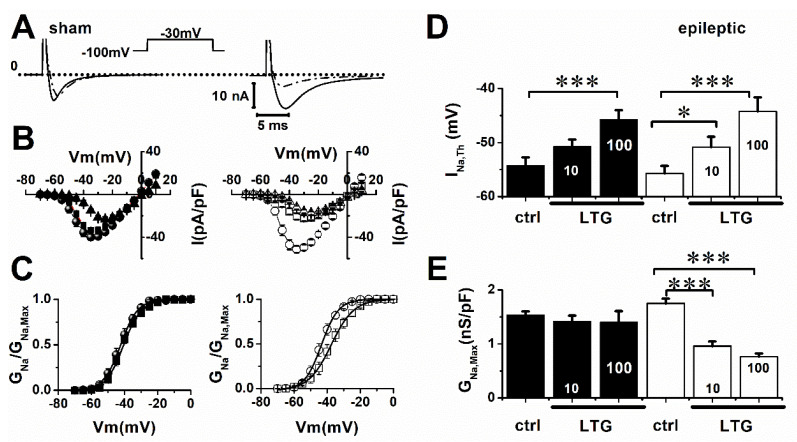
LTG blockade of I_Na_ is increased during epilepsy. (**A**) Inset: voltage clamp protocol. I_Na_ currents were activated by 25 ms voltage steps from −70 to 10 mV, in 5 mV increments, from a −100 mV holding potential. Representative I_Na_ recordings from sham and epileptic rat ventricular cardiomyocytes in control conditions (solid line), and in presence of 10 µmol/L of LTG (dash-dot line). (**B**) Current–voltage relationships (I/V) for sham (filled symbol) and epileptic (open) animals in control conditions (circle), 10 µmol/L (square) and 100 µmol/L of LTG (triangle). (**C**) Normalized I_Na_ activation curves for control and 10 µmol/L of LTG. Data points (±SEM) were fitted to a Boltzmann equation (solid lines). (**D**) LTG (10 µmol/L) depolarized I_Na_ voltage threshold (I_Na,Th_) only in epileptic animals, while 100 µmol/L of LTG significantly depolarized I_Na,Th_ in both groups. Each bar represents average values (±SEM) in control conditions and upon addition of 10 and 100 µmol/L of LTG. (**E**) LTG decreased Na^+^ channel conductance (G_Na,Max_) in epileptic animals. Average values of G_Na,Max_ in same conditions as (**D**). Tukey test using one-way ANOVA (ctrl vs. LTG), * indicates *p* ˂ 0.05, *** indicates *p* ˂ 0.001. Number of cells: Sham (ctrl, LTG): 13, 14. Epileptic (ctrl, LTG): 14, 13.

**Figure 2 pharmaceuticals-15-01208-f002:**
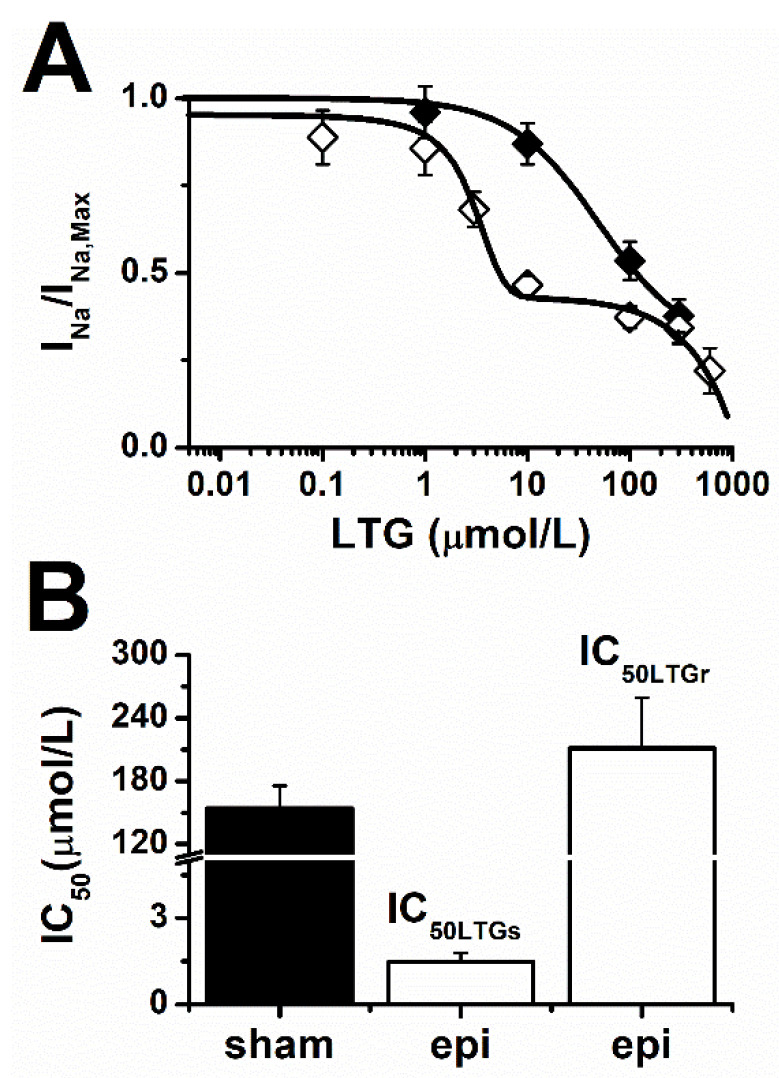
Epilepsy increased the affinity of LTG for I_Na_. (**A**) Dose response curves for sham (filled symbol) and epileptic (open symbols) animals. The curves were constructed by averaging the ratio between the I_Na_ control value (I_Na,Max_) and its amplitude at each LTG concentration tested. (**B**) IC_50_ values estimated by fitting single- and dual-dose response equations (solid lines) to sham (filled) and epileptic (open) animal data, respectively. Number of cells: Sham (ctrl, LTG): 7–14. Epileptic (ctrl, LTG): 8–13.

**Figure 3 pharmaceuticals-15-01208-f003:**
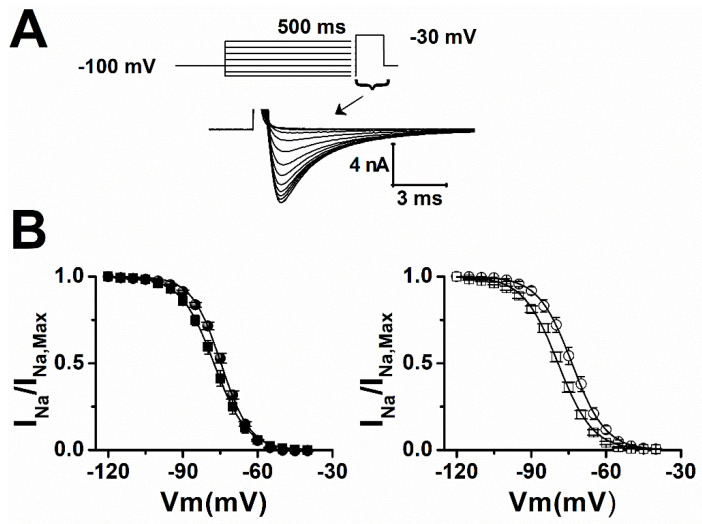
LTG hyperpolarized I_Na_ steady state-inactivation. (**A**) Representative I_Na_ inactivation recording obtained during a 15 ms test pulse to −30 mV following a series of 500 ms conditioning potentials from −120 to −40 mV in 5 mV increments (top). (**B**) Steady-state inactivation curves for sham (filled symbol) and epileptic (open symbol) animals in control conditions (circles) and 10 µmol/L of LTG (squares). Data points (±SEM) were fitted using a Boltzmann function (solid line). Number of animals: 3/group (ctrl and LTG). Number of cells: Sham (ctrl, LTG):15, 11; Epileptic (ctrl, LTG): 12, 19.

**Figure 4 pharmaceuticals-15-01208-f004:**
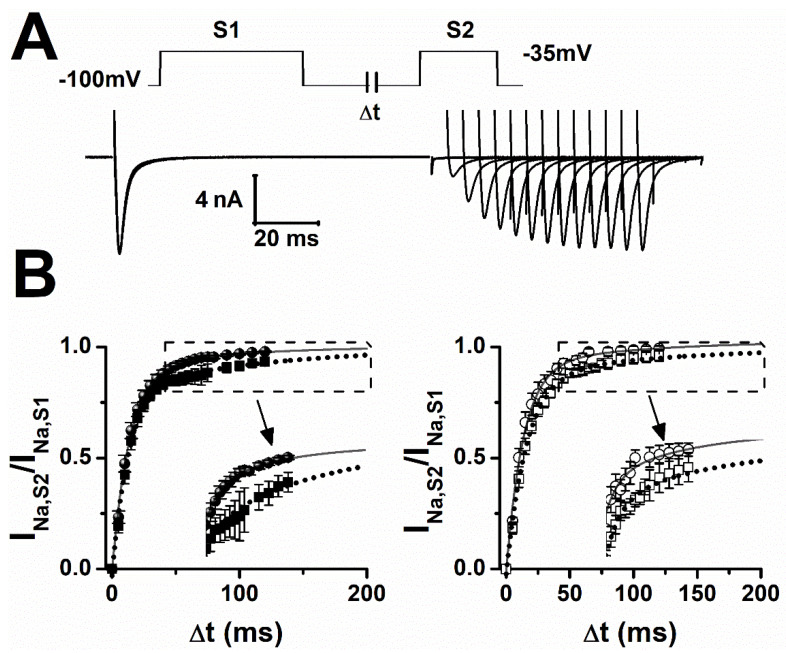
LTG slows I_Na_ recovery from inactivation. (**A**) Representative I_Na_ recording during recovery from inactivation. I_Na_ was elicited by two step pulses, S1 (100 ms) and S2 (20 ms) to −35 mV, separated by increasing intervals at the resting membrane potential, in increments of 5 ms. Holding membrane potential was −100 mV. (**B**) Recovery from inactivation curves for sham (filled symbol) and epileptic rats (open symbol) in control (circles), and 10 µmol/L of LTG (squares). Data (±SEM) were fitted to two exponential functions (solid: control, short-dot line: LTG) to estimate the time constants. Number of animals: 3/group (ctrl and LTG). Number of cells: Sham (ctrl, LTG):19, 9; Epileptic (ctrl, LTG): 9, 15.

**Figure 5 pharmaceuticals-15-01208-f005:**
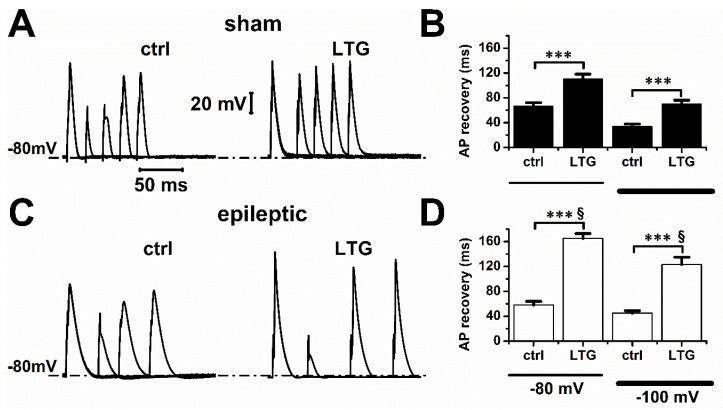
LTG prolonged cardiac action potential refractory period. (**A**,**C**) Representative action potential recordings from sham and epileptic rats in control (ctrl) and in presence of 1 µmol/L of LTG. Cardiomyocytes were held at membrane potentials of −80 or −100 mV and stimulated at a pacing cycle length of 2 ms (sham) and 10 ms (epileptic) for control experiments, and 5 ms (sham) and 50 ms (epileptic) to 1 µmol/L of LTG. For simplicity, only representative records of cells held at −80 mV are shown in the figure. However, as indicated above, the recordings were also performed on the same cell at −100 mV. (**B**,**D**) Action potential (AP) recovery time was measured as the time needed to recover a full amplitude action potential. Each bar represents the average value (±SEM). Tukey test using one-way ANOVA (ctrl vs. LTG), *** indicates *p* ˂ 0.001, and ^§^ indicates *p* ˂ 0.001 (sham vs. epileptic). Number of animals: 3/group (ctrl and LTG). Number of cells: Sham (ctrl, LTG):14, 14; Epileptic (ctrl, LTG): 14, 12.

**Figure 6 pharmaceuticals-15-01208-f006:**
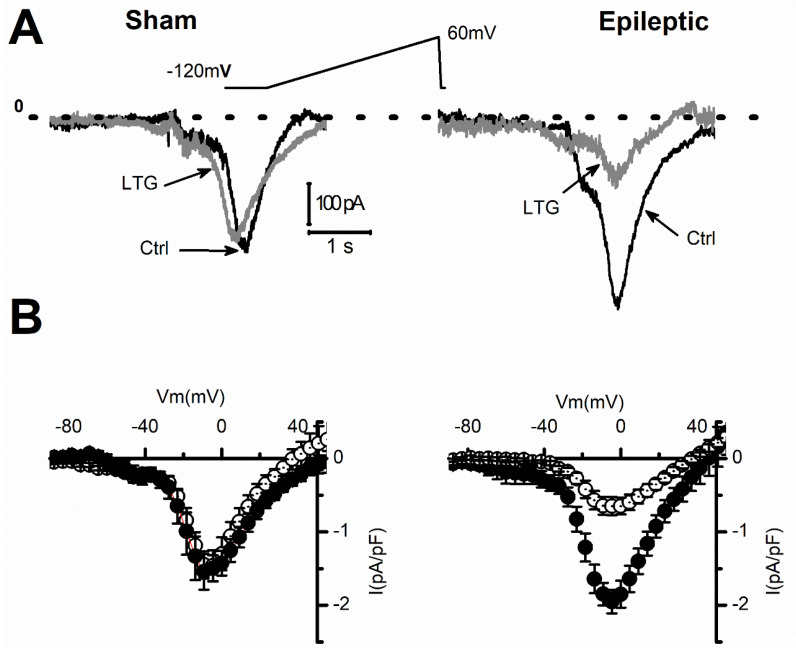
LTG blocked I_NaL_ current in cardiomyocytes from epileptic animals but had no effect in sham rats. (**A**) Representative I_NaL_ recordings from sham and epileptic rat cardiomyocytes in control conditions (ctrl, black) and in presence of 10 µmol/L/L of Lamotrigine (LTG, gray). Currents were elicited by a voltage ramp protocol from −120 to 60 mV with a 0.04 mV/ms rate. (**B**) IV-curves showing the average of current density values for sham (left) and epileptic animals (right) in control conditions (filled circles) and to 10 µmol/L/L of LTG (empty circles). Number of animals: 3/group (ctrl and LTG). Number of cells: Sham (ctrl, LTG):12, 12; Epileptic (ctrl, LTG): 14, 16.

**Table 1 pharmaceuticals-15-01208-t001:** Na^+^ channel activation parameters for control conditions and during exposure to 10 µmol/L of LTG. Tukey test using one-way ANOVA (ctrl vs. LTG), * indicates *p* ˂ 0.05.

Sham	V_0.5_ (mV)	Slope
ctrl	−42 ± 1	4.5 ± 0.2
LTG	−41 ± 1	5 ± 0.2
**epileptic**		
ctrl	−43 ± 1	4 ± 0.3
LTG	−38 ± 2 *	5.3 ± 0.2 *

**Table 2 pharmaceuticals-15-01208-t002:** Na^+^ channel inactivation parameters for control conditions and during exposure to 10 µmol/L of LTG. Tukey test using one-way ANOVA (ctrl vs. LTG), * indicates *p* ˂ 0.05, ** *p* ˂ 0.01.

Sham	V_h_ (mV)	Slope
ctrl	−74.5 ± 0.6	5.9 ± 0.2
LTG	−77.2 ± 1.2 *	6.5 ± 0.2 *
**epileptic**		
ctrl	−73.3 ± 1.4	6.5 ± 0.3
LTG	−80 ± 1 **	7 ± 0.2

**Table 3 pharmaceuticals-15-01208-t003:** Time constant values of I_Na_ recovery from inactivation. The relative weights for fast and slow components are indicated by I_f_ and I_s_, respectively. Tukey test using one-way ANOVA (ctrl vs. LTG), * indicates *p* ˂ 0.05, ** indicates *p* ˂ 0.01, and *** indicates *p* ˂ 0.001. ^§^ indicates *p* ˂ 0.05 (sham vs. epileptic).

Sham	τ_f_ (ms)	I_f_ (%)	τ_s_ (ms)	I_s_ (%)
ctrl	15 ± 2	92 ± 1	80 ± 2	9 ± 1
LTG	14 ± 1	80 ± 4 **	127 ± 3 ***	20 ± 1 **
**epileptic**				
ctrl	11 ± 1	87 ± 4	97 ± 8 ^§^	13 ± 4 ^§^
LTG	17 ± 2 *	88 ± 1 ^§^	130 ± 2 ***	12 ± 1 ^§^

## Data Availability

Data are contained within the article and Appendix A.

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
