# Peer review of "Epilepsy-Induced High Affinity Blockade of the Cardiac Sodium Current INa by Lamotrigine; A Potential for Acquired Arrythmias"

_pharmaceuticals, 2022, doi:10.3390/ph15101208_

Round 1

Reviewer 1 Report

This manuscript discusses Lamotrigine ( LTG, a pharmacologic agent that is used to treat bipolar disorder and epilepsy).  The authors studies in this manuscript its electrophysiological effects on ventricular myocytes by using patch-clamping techniques. Overall, the study is not clear in many sections including methods, results and discussion.

1- The experiment was run in 4 groups, Sham, Sham+LTG, epilepsy, epilepsy+LTG. Why is the data presented in 2 groups in Fig.1 ? all 4 groups need to be on the same graph and the data should be run for statistical analysis as a 2-way ANOVA. Did the authors use DMSO in the untreated control? since LTG is soluble in DMSO. what is the statistical significance of different groups at different voltages in the IV curve?

2- Before running the patch-clamping experiments, did the authors run simple ECG to check if the rats develop arrhythmia and how LTG affects the arrhythmia patterns?

3- Did the authors collect hearts and check for the expression of neuronal Na channel (proteins) in the heart and how epilepsy and LTG change the expression patterns?

4- Please compare all 4 groups together in Figure 2. 

5- Figure 5 and These statements " the reduction of APD occurs at 30% and 50% repolarization which is the region of major INa,L involvement" "Late sodium current contributes to AP repolarization plateau and its enhancement causes a longer QT interval" Where is late sodium current and how did the authors measure it? Please check PMID26092781. What is highlighted here is peak sodium current and not late sodium current?

6- If LTG affects the late sodium current, how the is APD90 not affected? Suppl. Figure 1

7- Methods: how was late current measured? which area of the sodium current trace was measured? What is the extracellular solution composition/concentration for recording the late sodium current? NaCl? the extracellular concentration measuring peak should be different from late sodium current.

8- What is the contribution of neuronal sodium channels to late sodium current in heart? Most of late sodium current comes from Nav1.5 in heart so if LTG affects the neuronal channels and the late current originated from the neuronal channels so the effects should be minimal. The data presented here is very confusing. 

Author Response

We would like to thank the reviewer for his comments and suggestions that help to improve the quality of this work. Our most sincere gratitude.
We have been asked to upload the revised version (in track changes mode) as well as the response to the reviewer. Therefore, you will find a merged pdf file with both documents.

Reviewer 2 Report

In the manuscript entitled “Epilepsy-induced high affinity blockade of the cardiac sodium current INa by lamotrigine; a potential for acquired arrythmias”, Contreras-Vite JA et al presented a biophysical characterization of the effects of LTG application on the sodium currents from the ventricular myocytes isolated from control and epileptic rats. The authors aimed to establish that the increased expression of neuronal sodium channel in the myocytes of the epileptic rats is one of the mechanisms for LTG-induced cardiac arrhythmia. The manuscript is written well, in a very condensed form. The manuscript would benefit from additional experiments to show the expression of the two types of channels (cardiac vs neuronal) in the myocytes of epileptic rats, to isolate them in electrophysiological experiments and to better obtain dose-response curves of LTG on each component in the native system.

The manuscript has no line numbers.

Introduction, paragraph 2: please provide the plasma concentration in mM too. 
Results, paragraph 1: Figure 1A displays representative traces and not quantitative analysis.
The statement “Current voltage relationships (Figure 1B) show that LTG blockade of INa saturated between 10 and 100μmol/L in epileptic animals” – the term saturated is not appropriate, since no dose response curve has been shown. 
Legend of Fig 1 A: the description of the inset is not corresponding to the figure.   Figure 1D shows only the shift of voltage threshold for 10μM LTG and not for the 100 μM LTG – please include the 3rd bar in the graph to match the text.
The legend of Figure
2: 2nd line, the voltage is -30 mV and not 30 mV.
Subchapter 2.1, “However, LTG effect on slow recovery was more important in cardiomyocytes from sham animals (Figure 3B, Table 3)” – is an overstated conclusion, not supported by the data: in Fig.3B, the two graphs are identical, and no statistical test is presented. It is not clear how are these effects more important for the control animals. Experimental proof is missing at this stage of data presentation. It appears that LTG has similar effects, the main differences are between epileptic and control rats.
“LTG increased refractory period by a factor of 1.9 for sham and 2.8 for epileptic animals.” – are these effects different? – mention please the statistical test used for comparison.
In general, the comparison between the effects of LTG in control and epileptic rats should be followed by a statistical test.

Experiments for Fig 4: how do authors justify the use of 1 μM and not of 10μM LTG.  
 According to the description of authors, Fig 5A the increase in the amplitude of the INaL is sufficient to explain the prolonged AP. Did authors try to apply another drug to block INaL or a nNaV to show that it has a similar effects in the same myocytes? Could other currents contribute to APD prolongation, such as Ca2+ or K+-currents? Could LTG affect these channels too in epileptic myocytes?
Why did the authors change the concentrations of LTG in the experiment? If 1μM LTG does not affect INa in control myocytes (according to Fig 6), what is the explanation for the reduction of APD30 and APD50 in these rats? (Fig S1).
How is the ratio of TTX sensitive vs resistant Na channels in control vs epileptic rats? Is it a switch of isoforms? Fig 6 is interesting and may show two components, but this figure should not be at the end, but in the beginning and it looks more like the trigger of the study and not the last experiment. Dose response curves on isolated Na currents would clarify the low affinity and the high affinity component.
The study would benefit from data showing the mRNA and protein level on nNaV vs Nav1.5 in the presented model.
Can the authors provide similar INa measurements from epileptic rats treated with LTG for a longer time?
Double check the abbreviations.

Methods: the authors did not mention if LTG was acutely or chronically applied, I assumed that it is an acute application, but how fast was the effect, is the action reversible`?
Is it a difference between control rats and sham rats?

Overall, the study is interesting, however, some more experiments would be required to establish the effects of LTG on the sodium channels in cardiac myocytes.

Author Response

(The authors gave the same response as above.)

Reviewer 3 Report

In this manuscript, authors provided an ionic basis for heart arrhythmia associated with an epileptic drug, lamotrigine (LTG), in an epileptic rat model. Building on the previous findings for the presence of neuronal type of Na+ channels in cardiac myocytes in rat and dog from the same group, the authors showed that LTG increased its affinity to Na current due to an epilepsy-induced increase in Na current, a depolarized shift of activation, a hyperpolarized shift of inactivation, and a slowed recovery from inactivation. These LTG-induced changes in Na+ channels provided the likely explanation for the Brugada syndrome and QRS-widening in clinical observations of epilepsy patients treated with LTG. 

Author Response

(The authors gave the same response as above.)

Round 2

Reviewer 1 Report

The authors responded to my previous comments and made changes to the manuscript accordingly.

Reviewer 2 Report

The response offered by Contreras-Vite and colleagues addressed many points. However, important issues remain to be addressed.

1. The separation of the two TTX-components to explain the effect of LTG,  wouId be an essential experiment. Authors should  repeat the DR curve for LTG in the presence of low dose of TTX. The main problem is that the authors did not really clarify if only Nav1.1 is increased or other TTX-sensitive alpha subunits like Nav1.2, Nav1.3, Nav1.6, Nav1.7 are also increased. IC50 for Nav1.1 is around 20 nM, and at this concentration all the TTX sensitive currents would be blocked totally or at least 50%. An alternative would be to obtain DR curves from HEK cells expressing  Nav1.x alpha-subunits.

2. In the previous article, the authors showed indeed that the protein levels of Nav1.1 and Nav1.5 were increased in this model for epilepsy. However, that experiment was performed for 4, or 3 rats. The article has been published in 2015. Were the data presented in the current manuscript recorded in the past from the same lot of rats used for Western blot? Even if we assume that there is little individual variability in the response to KA, wouldn`t authors push the study further to investigate the protein expression of other Nav1.x TTX-sensitive subunits? 

3. There is no doubt that INaL contributes to AP repolarization. However, Nav1.1 has a fast kinetic of inactivation, therefore how can the authors prove that INaL passes through Nav1.1 or any TTX-sensitive current and not through Nav1.5? Western blot showed increased Nav1.5 protein level. So it is not clear that LTG acts on the same current as low TTX to elicit the effects on APD.

Increased INaL is sufficient to induce APD prolongation, only if other mechanisms were investigated and they were unchanged. Moreover, in the 2015 article, the authors showed that even 1 nM TTX shortens APD90, while 1 microM LTG shortens only APD30 and AP50 and not APD90. Therefore other underlying currents may be involved. In general, when recording AP in cardiomyocytes,  it is possible to record the total outward current in the same experimental conditions. These data would have given some first answers to the levels of total outward K-current.

4. In the current article, also in the article used as reference, the authors have chosen to use several concentrations of the applied drug: LTG concentrations were of 1, 10, 100 microM, without a real consistency. It is not scientifically correct to compare experiments performed in different conditions. I understand that if the AP is not triggered in 10 microM LTG, but in 1 microM LTG, all experiments should be done with 1 microM LTG and after that try higher concentrations. If the AP starts from -80 mV, and the authors recognize the importance of the Vh for the availability of the channel, it should be possible to apply voltage protocols first from -120 mV and then from -80 mV in the same cell. Therefore, experiments are missing.

Minor.

L20, L60: the authors use the term "importantly" - Is this term more relevant than "more significant"? How important was this effect from the pathophysiological point of view? How was this shown? 

In conclusion, although the article shows the effect on INa in ventricular myocytes, the assumption that this effect is more significant on TTX-sensitive current was not well demonstrated.
